# Cultured meat platform developed through the structuring of edible microcarrier-derived microtissues with oleogel-based fat substitute

Feng-Chun Yen [1,2], Jovana Glusac [1,2], Shira Levi[1], Anton Zernov[1], Limor Baruch[1], Maya Davidovich-Pinhas [1] ✉, Ayelet Fishman [1] ✉ & Marcelle Machluf [1] ✉

With the increasing global demand for meat, cultured meat technologies are emerging, offering more sustainable solutions that aim to evade a future shortage of meat. Here, we demonstrate a cultured meat platform composed of edible microcarriers and an oleogel-based fat substitute. Scalable expansion of bovine mesenchymal stem cells on edible chitosan-collagen microcarriers is optimized to generate cellularized microtissues. In parallel, an oleogel system incorporated with plant protein is developed as a fat substitute, which is comparable to beef fat in appearance and texture. Combining the cellularized microtissues with the developed fat substitute, two types of cultured meat prototypes are introduced: layered cultured meat and burger-like cultured meat. While the layered prototype benefits enhanced stiffness, the burger-like prototype has a marbling meat-like appearance and a softer texture. Overall, this platform and the established technological basis may contribute to the development of different cultured meat products and promote their commercial production.

Cultured meat (CM) is considered a sustainable alternative to meat, providing meat-like structures with similar eating experiences to animal-derived meat[1]. Aiming for a low environmental impact, enhanced animal welfare, and an available protein source for the world's growing population, the development of CM products has galvanized profound interests in academia and industry with increased public acceptance[2]. However, for this field to reach its full potential, major challenges should be overcome, primarily the development of technological approaches for scalable cell expansion and cell-to-meat processing. For scalable cell expansion, the development of efficient expansion methodologies is crucial to generate vast amounts of cells that could be further processed into CM. The efficient expansion of anchorage-dependent cells in scalable bioreactors has been long

applied through the use of microcarriers on which cells attach and proliferate[3]. This culturing method offers the benefits of a large attachment surface area, nutrient homogeneity, higher growth rates, and culture scalability[4,5]. Commercially available microcarriers have been, therefore, used to support the growth of different cells using materials such as glass, polystyrene, gelatin, and collagen[6]. These microcarriers act as temporary scaffolds for the expansion stage, followed by costly cell harvesting steps that also reduce cell yield. Another approach, proposed in our previous work[7], is to use edible microcarriers for cell expansion in bioreactors, and then incorporate the entire cellularized microcarriers (microtissues) into the final CM product without requiring harvesting steps. This approach can also be used to enhance meat-like appearance, taste, and nutritional values,

[1]Faculty of Biotechnology & Food Engineering, Technion – Israel Institute of Technology, 3200003 Haifa, Israel. [2]These authors contributed equally: Feng-Chun Yen, Jovana Glusac. ✉e-mail: dmaya@bfe.technion.ac.il; afishman@technion.ac.il; machlufm@bfe.technion.ac.il

through the choice of the material from which the microcarriers are produced and through the encapsulation of nutraceuticals and coloring agents in the microcarriers.

Once the growing cells reach confluency on the edible cell microcarriers, the microtissues can serve as building blocks to produce various CM products through cell-to-meat processing approaches. Being incorporated into CM products, the cellularized microtissues should preferably contribute to the desired sensorial properties. Texture, or mouthfeel, which mostly relies on the stiffness and elasticity of the structures, can be adapted using different combinations of edible biopolymers, crosslinkers, and post-expansion approaches[8]. Cell-microcarrier interactions can also affect these properties. Hence, an additional static culture of the cellularized microtissues can induce aggregation, which is accompanied by the production of extracellular matrix (ECM) and protein deposition that may enhance structural stiffness[9]. Another cell-to-meat processing approach is homogenization. Homogenization destructs the structures of cellularized microtissues, thus preserving their cellular contents while avoiding the particulate texture and appearance. Further restructuring of the cellularized mass into stable, cohesive constructs can be achieved using enzymatic and physical approaches. Catalyzing crosslinking reactions, enzymes provide an effective and controllable manner for modulating the structural properties of food proteins[10]. When purified and added to food preparations, several enzymes that are approved as food additives, such as transglutaminase (TG), can improve the flavor, mouthfeel, digestibility, and nutritional values[11]. Physical processing approaches such as compression, cooling, and extrusion can be also applied, thus affecting the texture without altering the composition.

Another essential component of the meat that contributes to its tenderness and juiciness as well as to the overall taste is fat. Plant-based fat substitutes that resemble the sensorial and nutritional attributes of animal-derived fat have been introduced and tested for different food applications. Oleogelation, the process of giving solid properties to liquid edible oil by using different gelling agents, is a powerful approach that can aid the development of fat mimetics devoid of transfat, reduce saturated fat content, and offer health-promoting effects[12]. Oleogels can be formulated using a direct or indirect approach, where in the former the oil structuring agent is added directly to the oil, such as in the case of monoglycerides, waxes, etc.[13]. For the latter, additional processing steps are required due to the use of biopolymers such as proteins[14] and polysaccharides[15]. The combination of various structuring approaches can potentially improve the oleogel physical properties and nutritional values. More specifically, the incorporation of a protein-based shell to oleogel particles can facilitate their incorporation as a replacement for solid fat in CM.

In the current study, we, therefore, introduce a new approach for the development of CM products, based on edible microcarrier-derived microtissues in combination with oleogel-based fat substitutes (Fig. 1). We demonstrate the feasibility of scalable cell expansion on the edible microcarriers, incorporate fat substitute entities, suggest different complementary processing approaches, and thoroughly analyze the resulting CM prototypes in terms of their appearance, texture, and nutritional attributes.

## Results

### Scalable culturing system to produce cellularized microtissues

To achieve a scalable expansion of cells on our previously developed edible chitosan-collagen microcarriers[7], we used a 500 ml Applikon MiniBio bioreactor, which is considered a scalable system[16]. Using primary bovine mesenchymal stem cells (bMSCs, Supplementary Fig. 1), we studied the optimal culturing parameters to facilitate rapid cell growth, including the aeration method, stirring regimen, and medium exchange strategy in the bioreactor. As an aeration method, gas sparging was deemed unsuccessful since it generated foam, which disrupted the proper suspension of the microcarriers (Supplementary Fig. 2a and Supplementary Movie 1). A gas overlay that pumps air into the bioreactor's headspace, on the other hand, facilitated oxygen diffusion into the cell culture media without any negative effects, and was, therefore, selected for further use (Supplementary Fig. 2a).

In terms of stirring, two regimens were compared, where the stirring speed gradually increased from 45 rpm to 70 rpm or from 60 rpm to 80 rpm (Supplementary Fig. 2b). While the slower stirring speed allowed sufficient nutrient transfer and aeration, as concluded from the substantial cell viability (Supplementary Fig. 2c), it failed to adequately suspend the microcarriers. Thus, the seeded microcarriers had accumulated at the bottom of the vessel and formed aggregates (Supplementary Fig. 2d). Increasing the stirring speed from 60 rpm to 80 rpm, solved this problem, while still supporting cell growth (Supplementary Movie 2, Fig. 2a, b). Similarly, the medium exchange timeline was optimized, revealing that replacing 50% of the culture media on days 3, 5, and 7 leads to adequate nutrient renewal and waste removal.

These optimized culturing parameters were further applied to the bioreactor, and the scalable culture was compared to laboratory-scale culture conditions, i.e. suspension plate and spinner flask, representing static and dynamic cultures, respectively. A fluorescent live/dead cell staining and a quantitative cell viability test were used to image and quantify cells on the microtissues. The images clearly showed that bMSCs cultured in the scalable culture and the laboratory-scale controls gradually increased their confluency on the microcarriers,

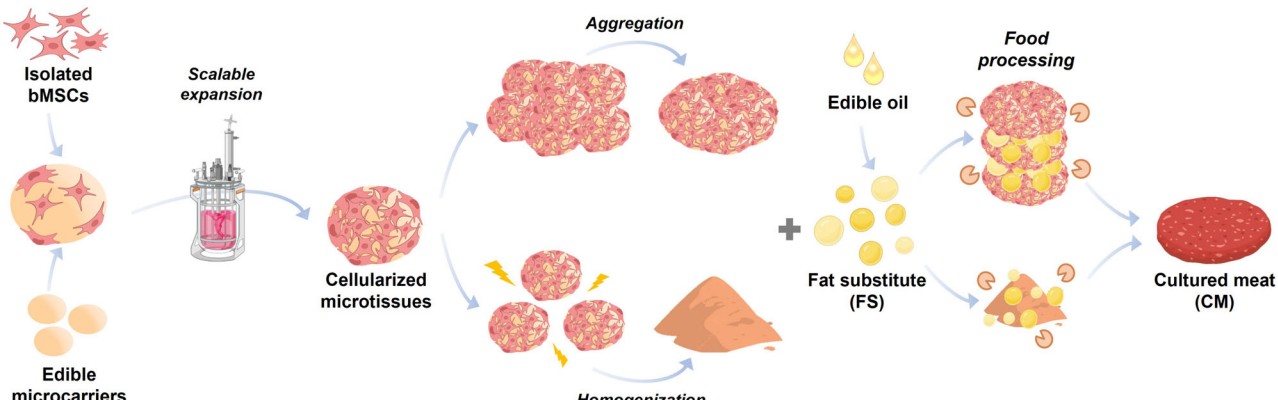

**Fig. 1 | Schematic illustration of the cultured meat platform.** Edible microcarrier-derived microtissues are first produced in a scalable bioreactor and then undergo processing approaches such as aggregation or homogenization. The processed cellular mass is further incorporated with an oleogel-based fat substitute followed by food processing methodologies to generate CM prototypes. Created with BioRender.com.

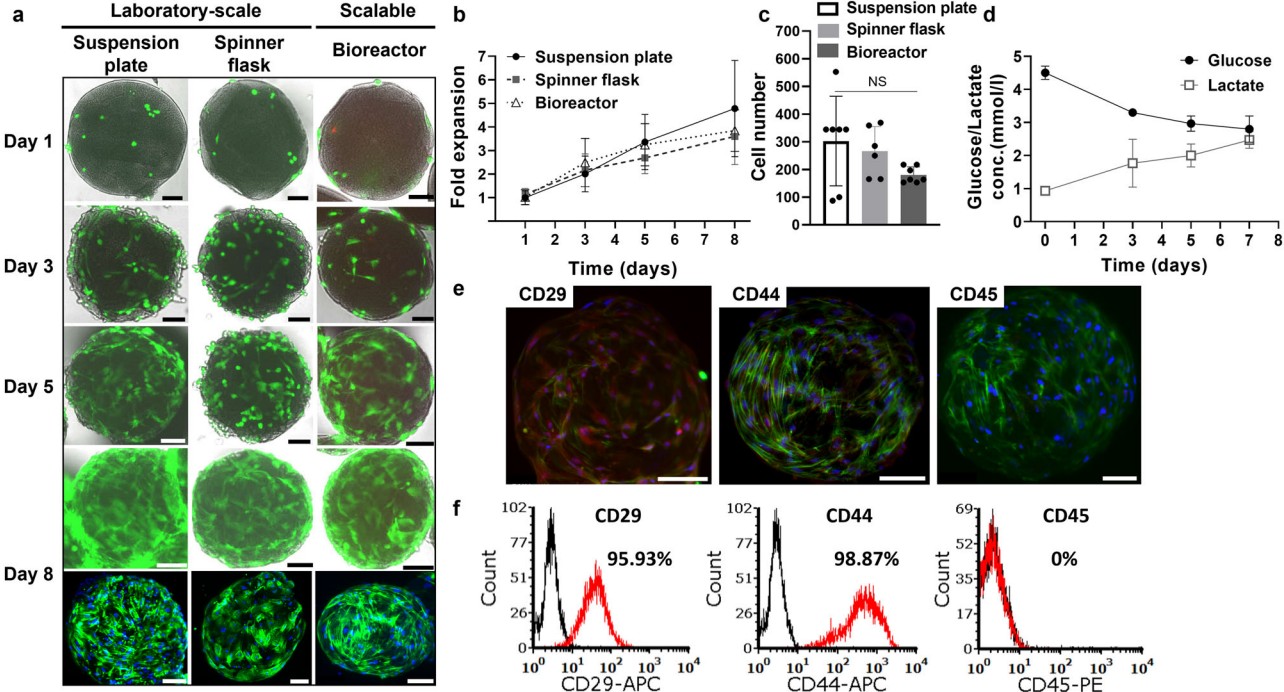

**Fig. 2 | bMSC expansion on edible microcarriers. a** Live-dead cell staining (representative images). Green: live cells (Fluorescein diacetate, FDA); Red: dead cells (Propidium iodide, PI). Bottom row - Green: Actin (Phalloidin); Blue: Nuclei (DAPI). Scale bars: 100 μm. **b** Fold expansion of bMSCs cultured on microcarriers along 8 days of expansion under laboratory-scale and scalable conditions. Results were normalized to the value of the suspension plate on day 1 and presented as mean ± s.d. ($n = 3$ independent experiments). **c** Number of cells per microtissues when cultured under different conditions presented as mean ± s.d. ($n = 6$ independent samples). NS: not statistically significant. Statistical comparison was performed by two-tailed t test. **d** Glucose and lactate concentration (mmol/L) of the cell culture medium in the bioreactor presented as mean ± s.d. ($n = 3$ independent experiments). **e, f** Immunophenotypic features of bMSCs following 8 days of expansion in the bioreactor. **e** Light-sheet fluorescence images of the cells stained for surface markers (positive: CD29, CD44; negative: CD45, representative images were from at least 4 independent samples with similar results). Red: Markers; Green: Actin (Phalloidin); Blue: Nuclei (DAPI). Scale bars: 100 μm. **f** FACS analysis of the cells using CD29, CD44, and CD45 surface markers. Source data are provided as a Source Data file.

reaching a complete coverage of the microcarrier by day 8 (Fig. 2a). Nevertheless, the static expansion within the suspension plate resulted in microtissue aggregation (Supplementary Fig. 3). The quantitative viability analyses further demonstrated that both scalable and laboratory-scale expansions reached the highest numbers of viable cells on day 8 (Fig. 2b). Cell expansion within the suspension plate, spinner flask, and bioreactor demonstrated similar cell growth with a comparable relative fold increase in cell number. Their calculated specific cell growth rates by day 8 of expansion were $0.22 ± 0.03$, $0.16 ± 0.04$, and $0.19 ± 0.05$ day$^{-1}$ in the suspension plate, spinner flask, and bioreactor, respectively with no statistically significant differences. These results were further validated through nuclei count, showing a similar amount of cell numbers per microtissue in all culturing conditions (Fig. 2c). Notably, a higher deviation of cell numbers per microtissue was observed in the suspension plate culture (Fig. 2c), while in the dynamic culture conditions (Spinner flask, Bioreactor) cell numbers were more homogeneous.

Throughout the bioreactor culture, we examined the glucose concentration in the cell culture media, which indicates the efficiency of cell growth. In addition, the lactate concentration was measured, aiming to detect amounts of waste metabolites produced from the cells. Both metabolite concentrations were measured before each medium exchange. The glucose concentration showed a gradual decrease from $4.5 ± 0.2$ mmol/L on day 0 to $2.8 ± 0.4$ mmol/L on day 7 (Fig. 2d). The lactate concentration showed a gradual increase from $0.9 ± 0.1$ mmol/L on day 0 to $2.5 ± 0.3$ mmol/L on day 7 (Fig. 2d), which is lower by an order of magnitude than the inhibitory concentration reported for human MSCs (35.4 mmol/L)[17].

As with any 3D culturing system, culturing cells on the edible microcarriers within a bioreactor can affect different aspects of their

behavior and growth. After verifying that a comparable growth rate and cell morphologies were obtained using this culturing system, we, therefore, studied the effect on cell behavior in terms of stemness. To this end, bMSCs' markers expression was investigated post-expansion using immunofluorescence staining that showed their specific surface markers were still positive for CD29 and CD44, while negative for CD45 (Fig. 2e). These results were further validated through flow cytometry, demonstrating the same immunophenotypes (Fig. 2f).

### Fat substitute production and characterization

The fat component in animal-derived meat greatly contributes to its taste, mouthfeel, and texture[18]. Aiming to mimic these properties of fat in CM, an oleogel-based fat substitute (FS) was developed. For its production, a combination of direct and indirect methods was used. Oil droplets were structured with glycerol monostearate (GMS) in a protein-aqueous solution using an emulsification procedure, followed by lyophilization (Fig. 3a). The first step in the production of the oleogel-based matrix was the generation of oleogel-in-water emulsion stabilized by protein where the oleogel particles are structured with GMS. Confocal laser scanning microscopy (CLSM) analyses of the emulsion showed dense and tightly packed oleogel droplets, with an average size of 1–3 μm (Fig. 3b). The thick protein layer around the oleogel droplets, visible at the highest magnification, suggests a stabilizing effect of the protein on the emulsion, which was also devoid of flocculation and coalescence. A random formation of protein aggregates was also observed in the bulk, characterized by a very intense blue signal, revealing the development of protein clusters. The second step in FS production was emulsion lyophilization. The morphology of the obtained FS was observed under polarized microscopy (PLM),

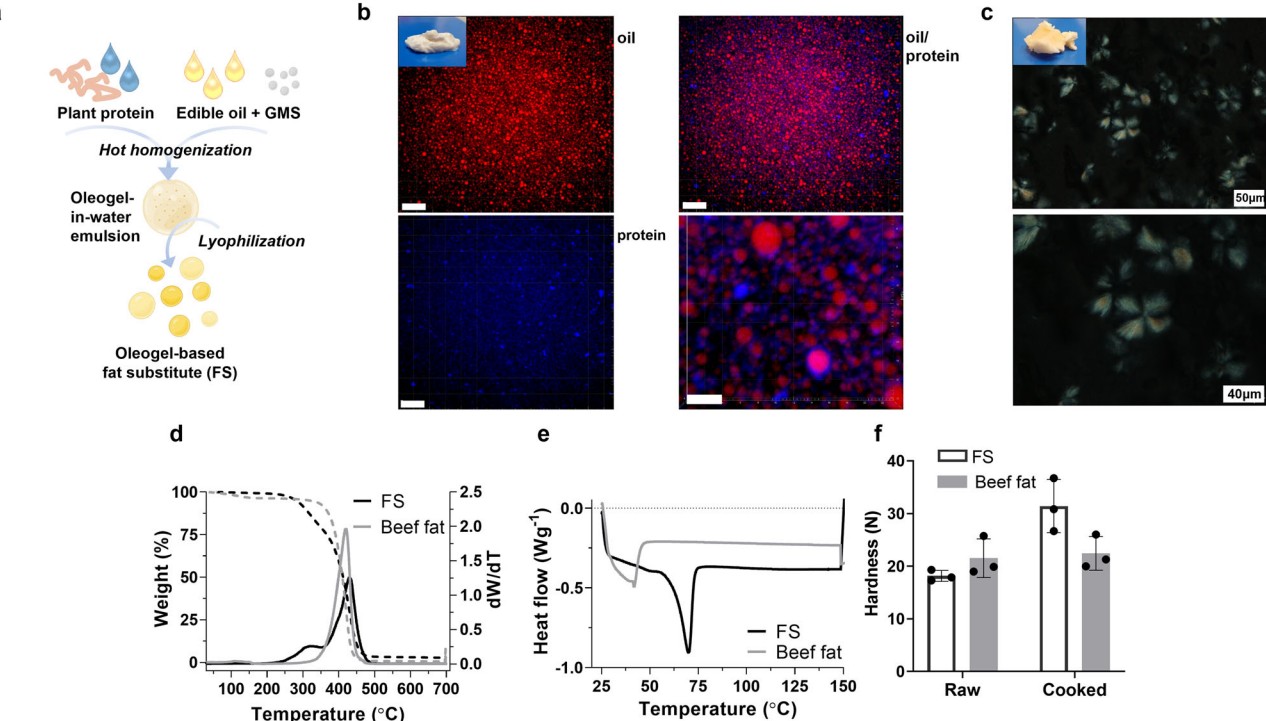

**Fig. 3 | Fat substitute production and characterization. a** FS production scheme based on o/w hot emulsification followed by lyophilization. **b** CLSM images of oleogel-in-water emulsion stabilized by protein. Oil, stained with Nile red, is seen in red, and protein, stained with Nile blue is seen in blue. Scale bars: 14 μm for all CLSM images, except the bottom right higher magnification image where the scale bar is 4 μm. Representative images are shown from 3 independent samples. **c** Polarized light microscopy images of FS. Scale bars: 50 μm (upper) and 40 μm (bottom). Representative images are shown from 3 independent experiments. **d** TGA curves of the FS and beef fat (left y-axis) and their derivative (right y-axis). **e** DSC thermogram (10 °C min⁻¹) of the FS and beef fat. **f** Hardness values of raw and cooked FS and beef fat presented as mean ± s.d. (*n* = 3 independent experiments). Source data are provided as a Source Data file.

demonstrating the organization of the GMS molecules into spherulite-like structures of approximately 50 μm diameter in the oil phase responsible for its solidification (Fig. 3c).

To address the properties of the produced FS, further characterization was performed in terms of thermal behavior (decomposition and melting), and textural properties, compared to commercial beef fat. The thermal decomposition of FS and beef adipose tissue was analyzed by thermal gravimetric analysis (TGA). Two weight-loss regions were observed in the FS; the first one between 250–350 °C associated with the volatilization and combustion of GMS (Supplementary Fig. 4a, b), and the second around 427 °C associated with the combustion of triacylglycerols (TAGs) comprising the canola oil (Fig. 3d and Supplementary Fig. 4a, b). The major weight loss region in beef fat had a peak centered at ~420 °C, thus overlapping the second peak of the FS which represents the TAGs combustion. Overall, derivative (dW/dT) profiles of FS and beef fat were very similar (Fig. 3d), indicating similar thermal behavior and composition. Melting properties evaluated by differential scanning calorimetry (DSC) revealed that both FS and beef fat was characterized by one major endothermic transition (Fig. 3e and Supplementary Fig. 4c). Beef fat showed a typical temperature peak (Tp) at ~42 °C[19] (Fig. 3e and Supplementary Fig. 4c). FS, on the other hand, was characterized by a sharp and narrow temperature peak at ~70 °C and 2-fold higher enthalpy compared to beef fat which can be related to the GMS melting behavior[20] (Fig. 3e and Supplementary Fig. 4c). Textural properties of raw and cooked FS were compared to raw and cooked beef fat, thus demonstrating similar hardness values (Fig. 3f), while the other textural parameters were less comparable (Supplementary Fig. 4d). In terms of appearance, FS had an overall similar appearance to beef fat, however, some differences were found in the measured color parameters (*L** (lightness), *a** (redness), and *b** (yellowness)) (Supplementary Fig. 4d, e). Particularly, higher values of yellowness were found in FS compared to beef fat, both in raw and cooked samples (Supplementary Fig. 4d, e), probably due to the presence of protein and the color of the original oil.

## Cultured meat prototypes: construction of microtissue aggregates

Advanced processing methods were tailored to fabricate different CM prototypes based on cellularized microtissues, oleogel-based fat substitute, and transglutaminase (TG)-crosslinking that facilitates the formation of a cohesive structure. One of the approaches that we addressed for microtissue-to-meat processing was microtissue aggregation. This approach aimed to strengthen the mechanical resistance of cellularized microtissues-based structures through cell-cell interconnections between cells on neighboring microtissues and ECM production, thus facilitating their integration with fat substitutes. To this end, two types of aggregates were generated: disc-like shaped and unorganized aggregates, through additional static culture post-expansion (Fig. 4a, b). This step allowed bMSCs to proliferate on the aggregates and reach the highest numbers of viable cells on day 7 (Fig. 4c). When comparing the stiffness of disc-like aggregates to microtissues, the aggregates presented around 40 folds with higher values of Young's modulus than the cellularized microtissues and the acellular microcarriers (Fig. 4d). Intriguingly, Young's modulus of the acellular microcarriers did not significantly increase after cellularization (Fig. 4d). To address a possible change in composition and, consequently, nutritional values, elemental analysis was performed. The aggregates presented higher N, C, H, and S contents than the microtissues, which presented higher N, C, H, and S contents than the non-cellularized microcarriers (Fig. 4e), likely due to the additional cell mass and the ECM deposited by the cells.

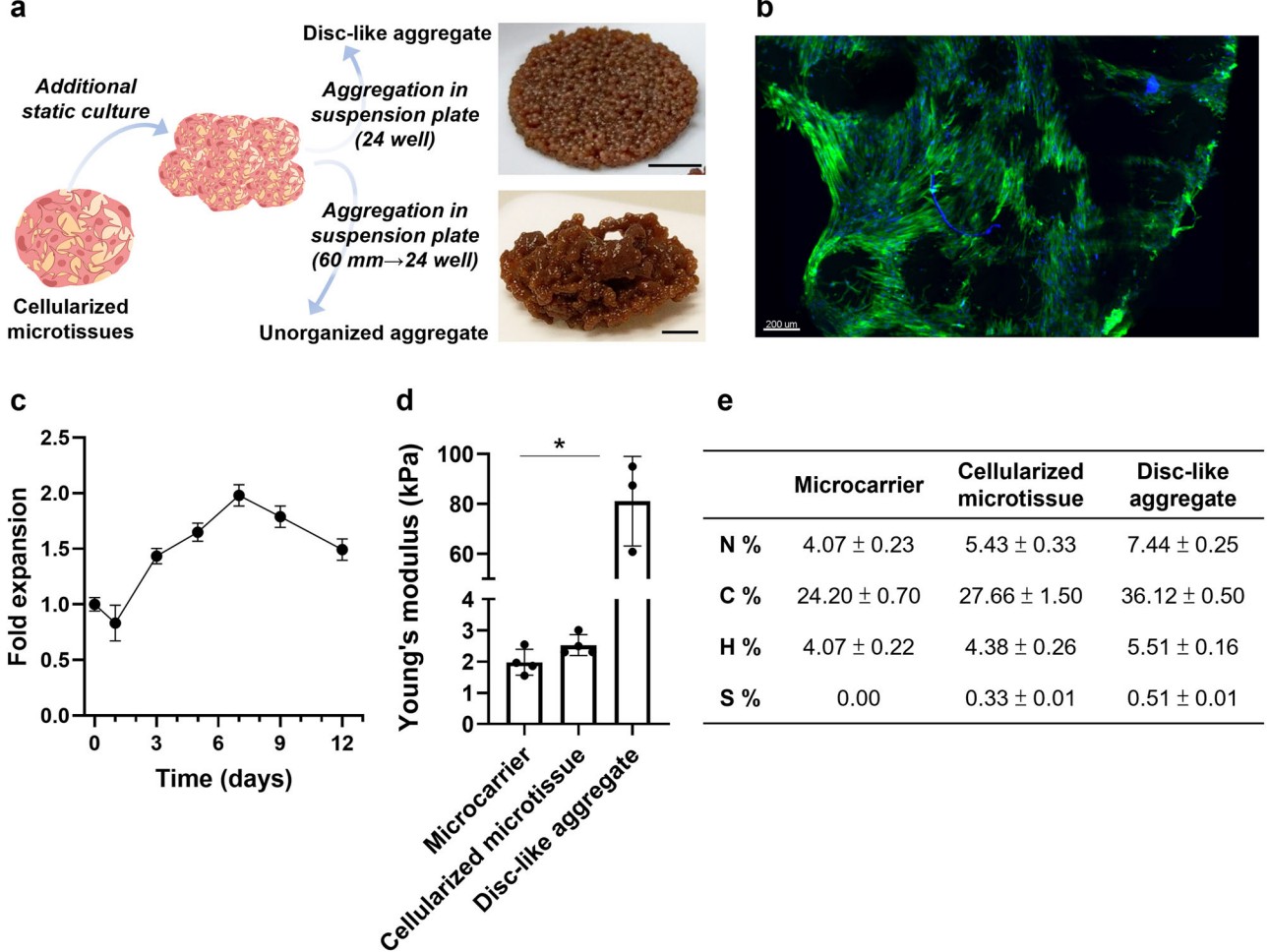

**Fig. 4 | Microtissue-derived aggregates. a** Production scheme of the disc-like and unorganized aggregates. Scale bars: 0.5 cm. **b** Light-sheet fluorescence imaging of the obtained disc-like aggregates. Green: Actin (Phalloidin); Blue: Nuclei (DAPI). Representative images were from at least 3 independent samples with similar results. **c** Fold expansion of cells cultured on the disc-like aggregates following 12 days of culture presented as mean ± s.d. ($n = 4$ independent samples). **d** Young's modulus of the microcarriers, cellularized microtissues, and disc-like aggregates presented as mean ± s.d. ($n = 3$ independent samples). Statistical comparison was performed by two-tailed t test, *$P = 0.0003$. **e** Elemental analysis of the microcarriers, cellularized microtissues, and disc-like aggregates. Source data are provided as a Source Data file.

## Cultured meat prototypes: structuring and characterization

The produced microtissue-derived aggregates were further integrated and structured into a unique CM prototype (Fig. 5a). Aiming to design a CM prototype with a marbling appearance, the layer-by-layer method was used for structuring, i.e. aggregate layer followed by a layer of FS added with TG. These components did not adhere to each other spontaneously or following pressure application, hence TG cross-linking was necessary to generate a cohesive structure (Fig. 5b and Supplementary Fig. 5a). Improved marbling appearance can be further achieved using microtissues aggregated into unorganized shapes, as seen in Supplementary Fig. 5b.

The second structuring approach aimed to design a CM prototype with a burger-like appearance, similar to ground meat. To this end, the microtissues were homogenized and then combined with FS and TG (Fig. 5a, b). The two CM prototypes were thoroughly characterized and compared to beef patties (Fig. 5c, d and Supplementary Fig. 6). The layered CM had moisture content similar to beef patties (~75%), while a lower moisture content of around 50% was measured in the burger-like prototype (Fig. 5c). The cooking loss of both CM prototypes was comparable to the beef patties, with ~30% weight loss (Fig. 5c). In addition, when cooking burger-like CM, the melting of the FS contributed to its juiciness and tenderness appearance (Supplementary Movie 3, Supplementary Fig. 5).

To gain better insight into the composition of CM prototypes and their similarity to beef patties, TGA was used. The first and most dominant stage, below 200 °C, is attributed to bound water loss and it is visible in all samples (Fig. 5d). The next degradation process is attributed to thermal dissociation of the quaternary protein structure with an onset above 229 °C ($T_{onset}$) in all samples (Fig. 5d and Supplementary Fig. 6). Interestingly, an increase of ~20 °C was observed in the temperature peak ($T_p$) of cellularized microtissues and aggregates compared to cell carriers alone, which can be attributed to the cellular component (Fig. 5d and Supplementary Fig. 6). Comparing the CM prototypes to beef patties, a similar decomposition profile is obtained, whereas the derivative curve reveals two peaks with a lower peak temperature in the first one and a higher peak temperature in the second (Fig. 5d and Supplementary Fig. 6), which is due to FS incorporation, as earlier shown (Fig. 3d). In terms of the fatty acid composition, the CM showed a similar profile to the FS, including the Omega 6 and Omega 3 fatty acids, with significantly lower content of saturated fatty acids, characteristic of the beef patties (Supplementary Fig. 7).

Next, the textural properties of raw and cooked CM prototypes were analyzed and compared to beef patties (Table 1). In contrast to the beef patties, the cooking did not enhance the hardness, adhesiveness, cohesiveness, and chewiness of the raw layered CM prototype (Table 1). The burger-like CM prototype had a more

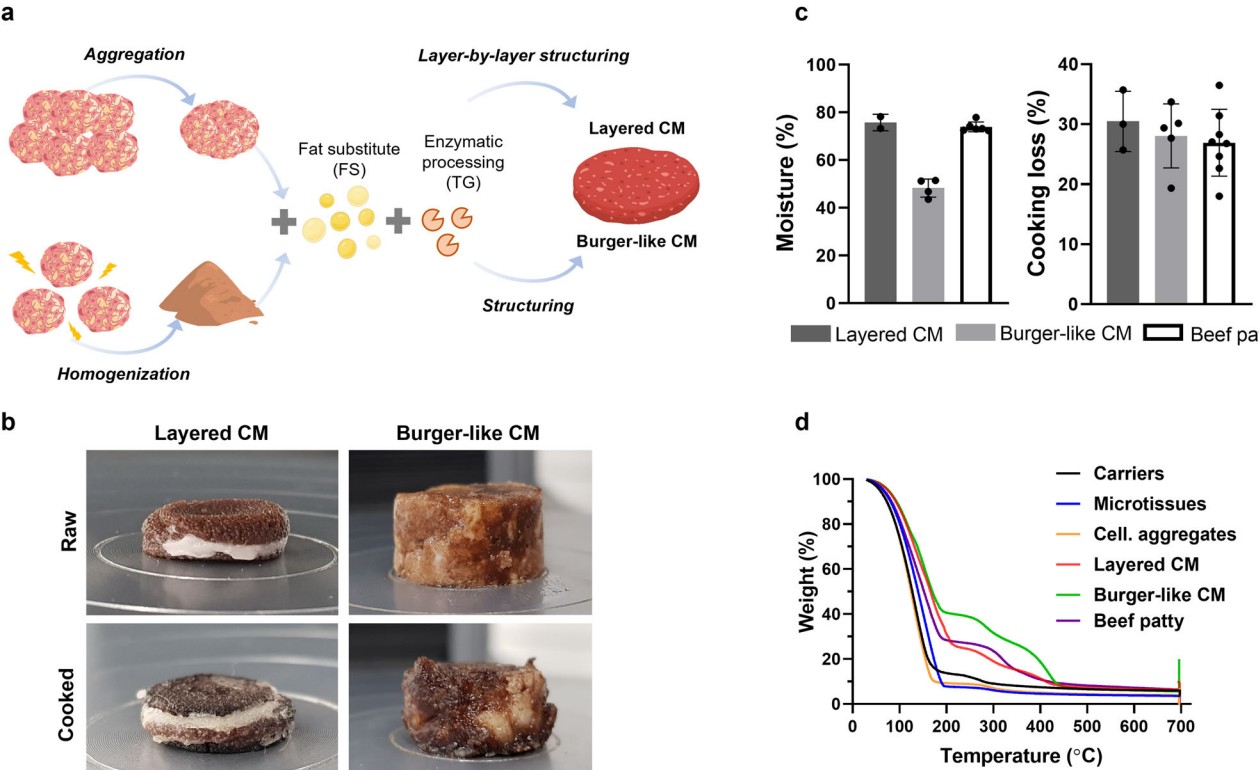

**Fig. 5 | Structuring and characterization of CM prototypes. a** Scheme of different CM prototypes structuring approaches. **b** The appearance of raw and cooked CM prototypes. **c** Moisture content (*n* = 6 for beef patties, *n* = 2 for layered CM, *n* = 4 for burger-like CM) and cooking loss (*n* = 8 for beef patties, *n* = 3 for layered CM, *n* = 5 for burger-like CM) in CM prototypes and beef patties presented as mean ± s.d. **d** TGA analysis presented as % weight against the temperature changes. Source data are provided as a Source Data file.

**Table 1 | Textural properties and color of raw and cooked CM prototypes and beef patties**

| | Layered CM | | Burger-like CM | | Beef patty | |
|---|---|---|---|---|---|---|
| | Raw | Cooked | Raw | Cooked | Raw | Cooked |
| **Textural parameters** | | | | | | |
| **Hardness (N)** | 6.43 ± 0.90 | 4.56 ± 1.09 | 1.64 ± 0.18 | 2.72 ± 0.41 | 3.76 ± 0.46 | 37.70 ± 3.98 |
| **Adhesiveness (Nmm)** | 2.78 ± 1.05 | 2.55 ± 3.74 | 0.97 ± 2.07 | 1.63 ± 0.51 | 0.56 ± 1.68 | 0.70 ± 2.37 |
| **Cohesiveness** | 0.56 ± 0.02 | 0.38 ± 0.17 | 0.24 ± 0.02 | 0.25 ± 0.02 | 0.40 ± 0.02 | 0.52 ± 0.02 |
| **Chewiness (N)** | 3.49 ± 0.64 | 1.17 ± 1.43 | 0.39 ± 0.08 | 0.65 ± 0.08 | 1.50 ± 0.14 | 18.10 ± 2.15 |
| **Springiness** | 0.97 ± 0.02 | 0.95 ± 0.09 | 0.98 ± 0.02 | 0.97 ± 0.01 | 0.99 ± 0.01 | 1.00 ± 0.00 |
| **Color** | | | | | | |
| **L\*** | 26.51 ± 1.38 | 33.93 ± 1.23 | 46.01 ± 1.68 | 30.92 ± 3.81 | 41.85 ± 3.21 | 49.84 ± 2.19 |
| **a\*** | 8.79 ± 0.92 | 5.07 ± 0.50 | 10.19 ± 0.12 | 8.91 ± 0.39 | 19.47 ± 1.86 | 9.69 ± 0.61 |
| **b\*** | 6.39 ± 1.08 | 0.98 ± 0.49 | 15.58 ± 0.89 | 5.68 ± 0.17 | 9.74 ± 1.27 | 9.79 ± 1.04 |

*L\*:* lightness, *a\*:* redness, and *b\*:* yellowness.

tender texture due to microtissue homogenization. Nevertheless, after cooking, a 1.7-fold increase in hardness was observed (Table 1).

Compared to the typical red color of raw beef, darker color in raw CM prototypes was observed (Fig. 5b, Table 1), as a consequence of EGCG crosslinking of microcarriers containing collagen and chitosan, as elaborated previously[5]. Further darkening is observed after cooking as a result of the Maillard reaction (Fig. 5b, Table 1).

## Discussion

CM aims to generate substantial cell mass through in vitro cell culture techniques and further structuring the cells into different meat analog products[2]. Our approach, proposed in the current work relies on three

principles: a) Scalable cell expansion using edible cell microcarriers, which supports the commercial viability of the production process through efficient cell expansion and the avoidance of costly cell harvesting steps. Furthermore, the use of edible microcarriers can contribute to the sensorial and nutritional properties of the CM product. b) Oleogel-based fat substitute, which mimics the appearance and mouthfeel of animal-derived fat with improved nutritional values, and c) Combination of biological and physical processing methods, which tailor the above building blocks into diverse CM products.

To achieve cell expansion in a manner that could be easily translated to a large-scale commercially-viable process, we demonstrated that bMSCs can successfully adhere and proliferate on edible

chitosan-collagen microcarriers when cultured in a scalable bioreactor. Furthermore, the growth kinetics obtained in the bioreactor were comparable to laboratory-scale expansions, thus providing a more relevant proof of principle for prospective industrial-scale production. The Applikon MiniBio bioreactor is considered scalable due to its geometry, which mimics the industrial-scale bioreactor, and the ability to translate the obtained shear forces and mass transfer coefficient into commercially-available industrial-scale bioreactors[16]. In general, the dynamic stirring in bioreactors facilitates enhanced oxygen and nutrient diffusion, which allows the use of larger vessels and homogenous conditions. For industrial-scale production, the homogeneity of the entire culture is of great significance to ensure product reproducibility and standardization, as also seen in our results, where the dynamic culture significantly improved culture homogeneity. Reaching confluent cell coverage of the microcarriers after only 8 days of expansion implies a potential for commercial feasibility, saving the expenses of media and equipment usage associated with longer growth periods. The typical bMSC markers remained unchanged postexpansion, indicating that the microcarrier-based expansion did not induce major cellular changes, and particularly, had managed to preserve the bMSC stemness.

Aiming to improve the structural integrity and stiffness of the cellularized microtissues, an additional culture of the cellularized microtissues was applied, to induce aggregation into disc-like and unorganized cellularized aggregates. Although the disc-like aggregates showed a fast fabrication and provided a higher stiffness, their structuring strategy relied mainly on layering approaches. Unorganized aggregates were shown to better mimic the marbling structures and appearances of the animal-derived meat. However, such structures were obtained after substantial additional culture, which involves additional costs and may limit their scalable production. Furthermore, a decrease in the viability of bMSCs grown on the disc-like aggregates was observed starting from day 7, suggesting that the cells did not have sufficient space to proliferate. Another possible explanation for this phenomenon is the limited oxygen and nutrient supply within the aggregates that trigger unavoidable cell death. Compared to the cellularized microtissues, the disc-like aggregates showed increased contents of N, C, H, and S, probably due to ECM deposition secreted from bMSCs. As collagen is a major component of the ECM[9], its presence provides an additional source of protein for CM[21].

To apply the aggregated or non-aggregated cellularized microtissues as the basis of CM products, we have further developed an oleogel-based FS that could enrich CM's sensorial and nutritional attributes. Using plant oil, we reduced more than 65% of the saturated fatty acid content compared to beef patties while greatly enhancing the quantity of Omega 6 and Omega 3 fatty acids. The formulation of the emulsion-templated oleogel-based FS was based on the gelation of oil by GMS along with the addition of protein, which stabilizes the oleogel particle surface. The main benefit of using GMS as an oleogelator is its recognition as a safe additive in food by the FDA[22] and the European Food Safety[23]. On the other hand, the addition of plant proteins that already have clean label status, can greatly improve the nutritional profile of the CM product and strengthen the interactions between the oleogel particles, forming a stable edible soft matter system[14,24]. While comprising higher protein content, the produced FS was comparable to beef fat in terms of appearance, color, thermal behavior, and hardness. Due to the addition of proteins, hardness was improved, while the presence of GMS affected thermal behavior and melting properties. The overall increased thermal stability accompanied by higher structural order compared to beef fat was largely due to the presence of GMS in FS, as previously reported in literature[25,26]. The higher melting temperature and melting enthalpy of FS compared to beef fat were supportive in preserving the tender texture of CM prototypes while cooking at 250 °C for 6 min. This is particularly obvious in the burger-like CM where the FS ratio was higher (30% w/w)

compared to layered CM (10% w/w), and where hardness increased after cooking. In general, in the case of cooked products, the FS must be able to replicate functionalities related to water and fat binding, thus providing juiciness and preventing water and fat loss during cooking[24,25]. Hence, strong intermolecular interactions between the proteins in the oleogel-based FS and proteins in the microtissues considerably enhance the CM integrity[12,14,24,26,27], which in turn leads to lower cooking loss, as observed in both CM prototypes. Another very important outcome of incorporating FS in CM prototypes is achieving a marbling appearance, which from the consumer point of view can translate to meat-like juiciness, tenderness, texture, and flavor. Therefore, based on thermal and textural properties while emphasizing the marbling effect, oleogel incorporated with plant protein is a promising alternative to be used in CM products. Additionally, higher protein content in the FS enhanced the nutritional composition compared to beef fat. A further advantage of the protein fraction in FS is its ability to bind microtissues through enzymatic crosslinking using TG and thus further accelerate the assembly of CM. Applying TG in CM processing was previously reported[28–30] and it can be assumed that via TG-crosslinking CM cohesiveness can be sustained, along with the improvement of other textural parameters.

Using the developed oleogel-based FS as an addition to the cellularized microtissues, we introduced here the manual assembly of two CM prototypes by using two processing approaches, e.g., cellularized aggregation (layer-by-layer approach), and structural homogenization. An advantage of the layer-by-layer approach when creating a CM product resembling a marbling appearance is its higher protein content, due to cells-secreted ECM. In terms of texture, this structuring method can lead to the development of a more delicate thick cut. Hence, it presents a good foundation for the muscle-like structure, and at the same time, high protein content can lead to an enhanced nutritional profile. The disadvantage of this approach, however, is the visible spherical shapes of the beads, which greatly devalue the appearance of the CM. Further optimizations can, therefore, meet this challenge while assembling microtissues with the FS into CM. For example, aggregation or fusion into a different structure than disc-like, such as the unorganized structure resembling a thick cut with a pronounced marbling appearance as shown in Supplementary Fig. 5b.

The second approach of structural homogenization led to CM with a burger-like appearance and structure. Although having a lower moisture content and a softer texture, as is often seen in cultured meat[28,30], a higher ratio of oleogel-based FS overall contributed to an appealing appearance in the raw and cooked CM. As can be seen from Supplementary Movie 3, when cooking the burger-like CM prototype, the melting of FS and water dehydration resemble real meat juiciness and tenderness appearance. Thus, this approach serves as a good platform for optimizing the "fat-to-meat" ratio while allowing the creation of marbling structure and appearance and making future processing of CM more feasible. Possible approaches to overcome the lower moisture content and hardness values obtained for this burgerlike CM prototype could be partial homogenization, the use of other homogenization techniques, or the application of further processing that will contribute to higher water holding capacity and texture improvement at the same time.

The nutritional composition of CM-based products has not been widely addressed thus far in previous reports or published research. Indirectly, we studied the composition of CM by TGA while elemental analysis was used for the analysis of the microcarriers, microtissues, and aggregates. An indication of the protein content can be obtained indirectly by measuring nitrogen, though it should be considered that a significant part of nitrogen content in the CM prototypes is attributed to the presence of non-proteinous chitosan in the cell carriers[7]. The overall profile of CM prototypes, obtained by TGA, was comparable to beef patties. However, further improvements should be applied to the prototypes to improve the nutritional profile.

To develop additional and diverse CM products, the two building blocks of this CM platform, i.e., microtissue and FS, can be further fine-tuned to control their size, mechanical, and structural attributes. Moreover, aggregates of different shapes and thicknesses as well as alternative crosslinking agents and additional processing methodologies can be applied, such as 3D printing thus tailoring the desired physicochemical properties of the final product, its structure, and shape[31]. Another possible approach is loading the oleogel-based FS with different macronutrients (proteins, fibers) and micronutrients (vitamins, minerals, antioxidants) to improve the nutritional profiles of CM products. Protein-based oleogels are relatively thermostable[14,24], which opens the possibility of tailoring the melting behavior through the type and quantity of oil structuring agent used. This will also influence the textural properties that together affect CM product quality and overall consumer acceptance.

To summarize, the current research demonstrates a CM platform based on cellularized microtissues and oleogel-based FS that are processed using different approaches to introduce two CM prototypes. By optimizing scalable methodologies to produce edible microcarriers-derived microtissues we suggest the feasibility of their commercial-scale production. As a fat substitute, we developed an oleogel system incorporated with protein, which presented comparable appearance, color, and hardness to beef fat with better nutritional values. Its combination with the cellularized microtissues enabled the engineering of two CM prototypes. The layered CM prototype was produced based on microtissue aggregates that supported better stiffness and nutritional values, while the burger-like CM utilized homogenized microtissues to imitate the marbling appearance of animal-derived meat. Altogether, in this work, we have established the technological basis for a unique CM platform that may broaden the applicability of CM products and accelerate their scalable production.

## Methods

### Production of edible cell microcarriers
Edible cell microcarriers were produced using a custom-made electrospray system as previously published[7]. Briefly, a solution of 2% chitosan (Glentham Life Sciences, GP8956) and 0.2–0.3% collagen (Sigma-Aldrich, C9879) was electrosprayed towards a sodium tripolyphosphate (TPP)-epigallocatechin gallate (EGCG) (Alfa Aesar, 013440 and Healthy Origins, respectively) crosslinking solution. The total volume of the microcarriers was calculated according to Eq. 1 while the total surface area of the microcarriers was calculated according to Eq. 2.

$$V = \frac{4}{3} \times \pi \times r^3 \times N \tag{1}$$

$$S = 4 \times \pi \times r^2 \times N \tag{2}$$

(V is the total volume of the microcarriers, S is the total surface area of the microcarriers, r is the average radius of microcarriers measured using ImageJ software (Fiji, version 1.53a, National Institute of Health, USA), and N is the total number of microcarriers calculated based on the ratio of the volume of sampled microcarriers to the volume of a single microcarrier).

### bMSC isolation and characterization
Bovine mesenchymal stem cells were isolated from a bovine umbilical cord taken during assisted delivery (Gazit dairy farm, Israel) as previously described[7]. Briefly, sterilized umbilical cord segments were treated for 30 min with 1 mg/ml Collagenase P (Sigma-Aldrich, 11-249-002-001) at 37 °C, and the digested tissue was seeded into cell culture plates. When reaching 80% confluency, the adherent cells were trypsinized for further passaging or cryopreservation. The cells were cultured in α-MEM (Gibco, 12000-063) supplemented with 10% fetal

bovine serum (FBS, Gibco, 10270-106), 1% penicillin/streptomycin (Sartorius, 03-031-1B), 0.8% amphotericin B (Gibco, 15290-026), and 0.25 ng/ml basic fibroblast growth factor (bFGF, PeproTech, 450-33) in a humidified incubator (37 °C, 5% CO$_2$).

The isolated cells were characterized for the presence of typical bMSCs cell surface markers using immunofluorescence staining and flow cytometry. Positive markers CD29 (BioLegend, 303008, dilution 1:200) and CD44 (BioLegend, 103011, dilution 1:200), as well as negative marker CD45 (Invitrogen, MA1-81458, dilution 1:100) were used for both analyses. Isotype control antibodies were: APC Mouse IgG1, κ (BioLegend, 400120, dilution 1:200), APC Rat IgG2b, κ (BD Biosciences, 553991, A95-1, dilution 1:200), and PE Mouse IgG1, κ (BioLegend, 400112, MOPC-21, dilution 1:200). Early passage (2–5) cells were quantitatively analyzed using flow cytometry (FACSCalibur™, BD Biosciences, USA) using the BD CellQuest Pro software (version 4.0.2) for data collection and the FCS Express Flow Cytometry software (version 7.16.0035, De Novo Software, USA) for data analysis. Qualitative results from immunofluorescence staining were obtained using a fluorescence microscope (Nikon, Japan). StemPro™ Differentiation Kits (Thermo Fisher Scientific, USA) were used to differentiate bMSCs (passage 2–3) towards chondrocytes, osteocytes, and adipocytes. Following the manufacturer's differentiation protocols, the differentiated cells were stained for Alcian Blue, Alizarin Red, and Oil Red O, respectively, and visualized under an upright binocular microscope (SZX16, Olympus, Japan). Representative images are shown (n = 4 independent samples).

### bMSCs expansion on microcarriers
**Cell seeding.** Edible cell microcarriers were seeded with the cells at a density of 8000 to 10,000 cells per cm$^2$ of microcarriers' surface area and incubated at 37 °C. After 24 h of incubation, the seeded microcarriers were stained with FDA (live cells) and PI (dead cells) to visualize the attached cells and their uniform distribution on the microcarriers using a fluorescence microscope (Nikon, Japan).

**Cell expansion.** Spinner flasks (1 L, Corning, USA) and an Applikon MiniBio bioreactor (500 ml, Applikon Biotechnology, Netherlands) were used for the dynamic cultures while 60 mm suspension plates were used for the static culture. Before inoculation of the seeded microcarriers, the spinner flasks and the bioreactor were treated with Sigmacote (Sigma-Aldrich, USA) to avoid cell adhesion on the surfaces of glass vessels during culture. For the bioreactor, the temperature sensor was calibrated using a 1-point calibration (0 °C), and the pH sensor was calibrated based on a 2-point approach (pH = 4, 7) before autoclaving. After autoclaving, the dissolved oxygen (DO) sensor was calibrated using a 2-point method. The bioreactor was first aerated with the air through its headspace for 15 min to set the first calibration point as 100% DO. The bioreactor was then pumped with nitrogen for 15 min to remove all the oxygen from the culture medium and set the second calibration point as 0% DO. Air was pumped as the gas source for the entire culture at 1 ml/min. Seeded cell microcarriers were then transferred with 200 ml culture medium to the bioreactor, where the temperature was set to 37 °C, pH to 7.2, and DO to 100%. The temperature, pH, and DO values of the bioreactor were automatically adjusted by the software to reach their set points over the entire culture. The following parameters were optimized for the scalable expansion of bMSCs in the bioreactor: aeration method, controlled stirring speed, and medium exchange strategy (Supplementary Fig. 2). For the spinner flask (200 ml working volume), the vessels were autoclaved before inoculation. Seeded cell microcarriers were transferred to the spinner flask with a 200 ml culture medium and kept in a humidified incubator (37 °C, 5% CO$_2$). For bMSC expansion in a static condition, seeded cell microcarriers (28.5 cm$^2$) were transferred with 10 ml culture medium to 60 mm suspension plates in a humidified incubator (37 °C, 5% CO$_2$).

Glucose and lactate concentrations (mmol/L) in the media were measured using a Roche Accutrend® Plus meter ($n = 3$ independent experiments). Cell viability was tested using AlamarBlue™ (Sigma-Aldrich, USA) and live/dead-cell imaging (FDA/PI, Sigma-Aldrich, USA) according to the manufacturer's protocol. Results are presented as mean ± s.d. ($n = 3$ independent experiments).

To follow cell growth, fold increase (FI) and specific growth rate ($\mu$, day$^{-1}$) were calculated according to Eqs. 3 and 4, respectively, where B/A is the ratio between cells on day 8 to cells on day 1, and $\triangle t$ is the time (7 days).

$$FI = \frac{B}{A} \tag{3}$$

$$\mu = \frac{\ln(\frac{B}{A})}{\triangle t} \tag{4}$$

### Characterization of cellularized microtissues

After 8 days of expansion, bMSCs on the cellularized microtissues were fixed with 4% paraformaldehyde (PFA, Electron Microscopy Sciences, 15710) for 20 min at room temperature and then washed with PBS. Cell morphology was visualized by staining the fixed cellularized microtissues with Phalloidin (actins, Sigma-Aldrich, P5282) and DAPI (nuclei, Biotium, 40011) using light-sheet fluorescence microscopy (Zeiss Z7, Germany). The generated 3D images were then used for the quantification of cell numbers through image analysis (Imaris, Oxford Instruments). Results were derived from at least 6 independent samples and presented as mean ± s.d.

Immunostaining analyses for bMSC markers used the same antibodies as described above for bMSC characterization and were visualized using a light-sheet fluorescence microscope (Zeiss Z7, Germany). The nutritional compositions of the microcarriers, cellularized microtissues, and disc-like aggregates were analyzed through elemental analysis using a CHNS Analyzer (Flash 2000, Thermo Fisher Scientific, USA). These samples (0.5 ± 0.1 g) were lyophilized to remove their water contents before measurements were taken. Mechanical properties were analyzed using a microscale mechanical test system (MicroTester MT G2, CellScale, Canada). Results of each measurement were derived from at least 3 independent samples and presented as mean ± s.d.

### Microtissue aggregation

For the aggregation of the cellularized microtissues, they were transferred into suspension plates and statically cultured. The disc-like aggregates were statically cultured in a 24-well suspension plate for 7 days, while the unorganized aggregates were statically cultured in a 60 mm dish for 7 days and then transferred to a 24-well suspension plate for additional 7-day culture. 50% of the medium was changed every other day. AlamarBlue™ viability tests were performed along 12 days of static culture in a 24-well suspension plate ($n = 4$ independent samples).

### Oleogel-based fat substitute production

For oleogel-based FS production, the method of oleogel-in-water emulsion-based template was used[16–19]. Chickpea protein dispersion (4.5%, w/w) in distilled water was stirred overnight at 4 °C. The oleogels were prepared by using canola oil (20% o/w, w/w) followed by the addition of a 20% (w/w, out of the oil phase) GMS (DMS 0091, Palsgaard®, Denmark). Emulsions were obtained by shear homogenization (16000 rpm/ 3 min). After homogenization, emulsions were gradually cooled (RT for 4 h, 4 °C for 4 h, and −80 °C overnight) and lyophilized. Altogether, the developed oleogel-based FS comprised 68% canola oil, 15% chickpea protein, and 17% GMS. The FS was compared to commercial beef fat (purchased from a local butcher shop).

### CM prototypes assembling

The cellularized microtissues were assembled into CM prototypes following two processing approaches 1) cellularized aggregation, and 2) homogenization as presented in Fig. 5a, along with oleogel-based FS and TG addition.

The aggregated cellularized microtissues were structured using a layer-by-layer approach to resemble a marbling appearance (Fig. 5a). Three layers of disc-like aggregates were used for structuring one layered CM prototype with 0.8 cm diameter and 0.4 cm thickness. Transglutaminase (ACTIVA®TG-TI, 1% in maltodextrin, activity: ~100 U/g, Ajinomoto Food Ingredients LLC., Chicago) was dispersed in phosphate buffer saline (PBS, 1:1 w/w) and mixed with FS (3:2 w/w). Between each layer of disc-like aggregates, a thin layer of FS + TG paste was applied. The composition of the final layered CM consisted of 75% w/w cellularized aggregates, 10% w/w FS, and 15% w/w TG in PBS. Layered CM was pressed with a syringe plunger and stored at 4 °C (Supplementary Table 1).

The second approach aiming for CM prototypes resembling burger-like appearance, comprised structural homogenization of microtissues, followed by FS and TG addition (Fig. 5b). Cellularized microtissues were homogenized with Precellys 24 tissue homogenizer (Bertin Instruments, France) for 5 s at 4000 rpm and mixed with TG (7.5:1). A mixture of homogenized microtissues along with TG was then gently mixed with FS. The composition of one final burger-like CM prototype consisted of 60% w/w cellularized aggregates, 32% w/w FS, and 8% w/w TG. Molds (2 cm diameter) were filled with the mixture, pressed with a syringe plunger, and stored at 4 °C (Supplementary Table 1).

As a reference, beef patties were prepared from fresh lean beef (4.3% fat) that was purchased from a local supermarket and structured into patties as in the burger-like CM prototype.

### Moisture content

Beef patties and CM prototypes were weighed and lyophilized. Moisture content was determined according to Eq. 5. Results are presented as mean ± s.d. (n = 6 for beef patties, n = 2 for layered CM, n = 4 for burger-like CM).

$$\text{Moisture content}(\%) = \frac{\text{Raw weight(g)} - \text{Lyophilized weight(g)}}{\text{Raw weight(g)}} \times 100 \tag{5}$$

### Cooking loss

Due to the different sizes of CM prototypes, samples were cooked on an electric hot plate under slightly different conditions. Thus, layered CM was fried at 150 °C for 5 min and 250 °C for 1 min, followed by cooling to 25 °C[23]. The burger-like CM prototype was fried at 250 °C for 6 min, as well as the beef patties (and FS and beef fat). Cooking loss (CL) corresponds to the weight of the CM prototypes (n = 3 for layered CM, n = 5 for burger-like CM) and beef patties (n = 8) before and after cooking, which was expressed as a percentage and calculated according to Eq. 6.

$$\text{CL}(\%) = \frac{\text{Raw weight(g)} - \text{Cooked weight(g)}}{\text{Raw weight(g)}} \times 100 \tag{6}$$

### Textural properties

Samples were analyzed as follows using a Texture Analyzer (TA1, Lloyd Instruments, UK) ($n = 3$ independent experiments). The samples were compressed by a stainless-steel plate (11.5 cm in diameter) twice to 50% of the initial height at a constant speed of 50 mm/min with a 10 N or 50 N load transducer. The burger-like CM, beef patties, and beef fat

samples were 1.8–2 cm in diameter and 2 cm in thickness, while the layered CM was 0.8 cm in diameter and 0.4 cm in thickness.

The following texture parameters were obtained: hardness at 50% of deformation, adhesiveness, cohesiveness, and chewiness. Hardness was defined by peak force (N) during the first compression cycle. Cohesiveness was calculated as the ratio of the area under the second curve to the area under the first curve (dimensionless). Springiness was defined as the distance over which the material recovers its height between the end of the first bite and the start of the second bite. Adhesiveness is the work required to overcome the adhesion force generated between food surfaces and substances with which the food comes into contact. Chewiness (the parameter used to describe solid food) was obtained by multiplying hardness, cohesiveness, and springiness. All parameters were calculated by the NexyGen Lloyd software version 4.1 (Lloyd Instruments Ltd., UK).

## Thermal analysis
The melting properties of FS were analyzed using differential scanning calorimetry (DSC) by using DSC 250 apparatus (TA Instruments, New Castle, DE, USA). Samples 3–5 mg ($n = 4$ independent experiments) were weighed in aluminum pans and heated from 25 to 150 °C at a rate of 10 °C min$^{-1}$ under nitrogen at a flow rate of 50 ml min$^{-1}$. As a reference, an empty pan was used. Melting properties were analyzed and compared to the melting properties of beef fat and GMS, by using TRIOS TA Universal Analysis Software (TA Instruments, version 5.1.1.46572).

Thermal gravimetric analysis (TGA) data were obtained using Discovery TGA 5500 (TA Instruments, New Castle, DE, USA) under nitrogen at a heating rate of 10 °C min$^{-1}$, monitoring weight loss as a function of temperature, until a maximum temperature of 700 °C was reached. Samples ($n = 3$ independent experiments) were analyzed using TRIOS TA Universal Analysis Software (TA Instruments, version 5.1.0.46403) and the results were expressed as a weight loss curve and its first derivative[20,21].

## Instrumental color measurement
The color was measured using a Minolta Chroma Meter CR-300 colorimeter (Minolta, Ramsey, NJ), calibrated with one standard tile – white using illuminant D65. Results ($n = 3$ independent experiments) were expressed as $L^*$ (lightness), $a^*$ (redness), and $b^*$ (yellowness) in terms of the CIELab system and were instrumentally calculated from the Minolta spectral curve[32].

## Spinning disk confocal microscopy (SDCM)
Spinning Disk Confocal microscope (Nikon, Yokogawa, Japan W1 Spinning Disk) was utilized to study the microstructure of the o/w emulsions. The emulsions were dyed before production with 0.1 % w/v Nile red (Sigma-Aldrich, N3013) and 0.1 % w/v Nile blue (Sigma-Aldrich, N0766) or with both dyes. The emulsion was placed on a glass microscope slide and covered. Images were processed by Imaris 9.0.2 software (Imaris, Oxford Instruments). Representative images were shown ($n = 3$ independent samples).

## Polarized light microscopy (PLM)
The microstructural analysis of FS was performed using a light microscope (BX51P, Olympus, Japan) in the bright field and polarized mode. Images were taken at RT using an Olympus DP71 digital camera. Representative images were shown ($n = 3$ independent experiments).

## Fatty acids profile
Soxhlet fat extraction was used to isolate the fat content of the samples. The lyophilized samples were weighted and placed in the thimbles and the extraction was done by adding petroleum ether (boiling point 40–60 °C) (Bio-Lab Ltd., 001718050100) to the thimbles in the Soxhlet extractors for 1 h of heating. The fatty acid profiles of the samples were then generated by gas chromatography with flame-ionization detection (GC-FID) of the esterified fatty acids, as described earlier[33].

## Statistical analysis
Statistical analysis was performed using GraphPad Prism 8 (version 8.4.3, San Diego, USA) software. ANOVA was used when more than two conditions were compared with each other and for only two conditions; the pairwise t-test comparison was performed. In addition, a classic one-way ANOVA was performed, with the Holm-Šidák post-test for multiple comparisons. All error bars represent standard deviation (SD). $p$ values were considered significantly different when $p < 0.05$. When differences were not significant, no marks were added to the graphs.

## Reporting summary
Further information on research design is available in the Nature Portfolio Reporting Summary linked to this article.

# Data availability
All relevant data supporting the key findings of this study are available within the article and its Supplementary Information files or from the corresponding author upon reasonable request. Source data are provided in this paper.

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

## Acknowledgements
We acknowledge The Good Food Institute and The Russell-Berrie Nanotechnology Institute (RBNI) at the Technion for supporting this research. F.C.Y. is the recipient of the Israel Ministry of Foreign Affairs Scholarship for international students. J.G. is the recipient of the Lady Davis Fellowship. Schemes were created with BioRender.com under a full license.

## Author contributions
M.M., M.D.P., and A.F. conceived, designed, and supervised the research. F.C.Y. and J.G performed the research with the help of S.L. and A.Z. F.C.Y., J.G., and L.B. analyzed and interpreted the data with the help of M.M., M.D.P., and A.F. F.C.Y., J.G., L.B., M.M., M.D.P., and A.F. prepared the manuscript.

## Competing interests
The authors declare the following competing interests: MM is a shareholder in Meatafora Ltd. The other authors declare no competing interests.
