## [Peer Review File · Nature Communications]

REVIEWER COMMENTS

Reviewer #1 (Remarks to the Author):

This work aims at the development of distinct cultured-meat prototypes, using oil gelation techniques. The author's approach is fairly interesting and the overall methodologies and strategies reported are suitable. This work indeed relevant to the science field advancement and therefore of interest to Nature Communications Journal.

Some methodologies details are missing and is pointed out for the authors to address in the revision. I would like to see some aspects regarding nutrition addressed in the work since, as the authors pointed out, these approaches could be important for the tailoring aspect of the produced systems not only concerning the mechanical and texture properties but also for the nutritional aspects as well. There are some concerns (below) that the authors can address to improve the manuscript.

-Abstract-

In my opinion, lacks objectivity and should be improved in that aspect. The main findings should be highlighted and well described (it seems a collection of bullet points).

"..exhibiting a similar..."

"...marbling appearance and a softer..."

Introduction

"...alternative to..."

Can the authors elaborate on this

"CM product to enhance meat-like appearance, taste, and nutritional values without requiring harvesting steps." In what way the nutritional values are affected and the feasibility of it.

When confluency is reached in what way the reactor volume limits the yield of the whole process?
There is the possibility to promote confluency in several layers?

Why not consider adipocyte growth as a complement for fat production?

-Results-

Is Minibio a trademark? If so please revise the wording.

The foaming was a sole consequence of using the sparger or the agitation was responsible for it?
Would a bubble column type-reactor with a bottom gas (porous or needle) sparger work? That would possibly avoid the microcarriers aggregation and foster a good enough gas dispersibility.

Figure 2 should be revised to improve readability. Font size must be increased, as well as micrographs size. There is no readability in Figure 2F. The authors should consider dividing the image.

Was there any statistical difference among the cell number when the culture was performed under different conditions?

"To grant CM with the sensorial and nutritional properties attributed to the fat component of animal-derived meat, an oleogel-based fat substitute (FS) was developed." this sentence is confusing to me. Please rewrite it.

It is not clear why TG was needed to produce CM prototypes. Can the author elaborate on that?

In figure 4A, the addition of a scale bar in the aggregates pictures would be helpful.

Correct the yy axis range in Figure 5d. The way it is portrayed (starting at 40%) is misleading and incorrect.

What was the fat content of lean beef?

Why there is no data on the textural properties of beef?

Why microcarriers can contribute to the sensorial and nutritional properties?

Aiming at increasing the scale of microcarrier production, are expected major cellular changes or even aggregation-related problems?

"Hence, strong intermolecular interactions between the proteins in the oleogel-based..." substitute "strongly" with another word to avoid repetition.

Any thoughts on how to tackle the "dark coloration" issue?

Fatty acid composition data for the CM prototypes would be a relevant result that would be beneficial to the overall discussion. Is it possible to add that?

-MM-

In the oleogel production description, the first sentence (regarding lyophilisation) must come at the end of the paragraph.

Briefly describe how moisture content was determined.

For textural tests define the probe dimensions as well as the dimensions of the tested samples.

Did the authors perform some cutting tests? That would be relevant for a better understanding of the internal structuring of the prototypes.

Reviewer #2 (Remarks to the Author):

Comments to authors: Cultured meat prototypes developed through the structuring of edible microcarrier-derived microtissues with the oleogel-based fat substitute

The authors fabricated different meat analogues by integrating edible microcarriers (Chitosan – Collagen) and oleogel-based fat substitutes (FS). This can be done because of cell culture parameter optimisation in both static and dynamic (spinner flask and bioreactor) conditions which demonstrate the upscaling feasibility. This approach showed that the authors were able to fabricate the meat analogue in different formats, from layer-by-layer mixed meat and fat structure to burger-like structure. Moreover, the authors also explored food characteristics, including moisture, nutritional/elemental composition, texture, cooking suitability due to thermal degradation and meat's colour. This is a resonable work that incorporates both plant-based fat substitutes and the confluence cells on micro-carriers. The authors pointed out that sustainability is one of the main reasons behind the attention on cultured meat.

Although chickpea protein and canola oil are non-animal based, chitosan and collagen are still derived from animal sources. They are still being used to make edible micro-carriers which is contradictory. Regardless, the paper is a proof of concept for using a combination of micro-carriers and FS for manufacturing cultured meat. This work still contains some issues which need to be addressed. Please refer to the comments below.

Major comments:

1) Interestingly, the authors are able to differentiate bMSCs in three different cells, which are chondrocytes, adipocytes, and osteocytes. However, the dominant cells in cultured meat (CM) should be myocytes (muscle cells). The protein in CM should come from the differentiated myocytes, which are myoblasts. It is well known that the differentiation of myoblasts into myotubes and, followingly, myofibers from MSCs is essential to obtain the desired texture of the final culture meat product. Could authors deliberate more on why the authors did not try to differentiate bMSCs into myocytes or do any analysis of this cell type?

2) The paper mainly focused on the food science/processing and cooking aspects and failed to represent the molecular aspects of muscle differentiation during the development of the prototype.

3) Elemental analysis may not be the most appropriate for understanding the nutritional composition of the fat substitutes. Could the authors elaborate more on the choice of the method instead of considering standardised nutritional analysis methods?

4) On page 13, Figure 4e: The authors performed an elemental analysis of the cellularised microtissues and disc-like aggregates. However, it is necessary to compare this with decellularised microtissues. This is because the edible micro-carriers also contain the protein element from collagen. Thus, N and S elements which are present more in protein than fats, may come from the micro-carriers themselves but not from the cells since bMSC cells have not differentiated into muscle cells.

5) Why did the authors compare both CM prototypes with lean beef rather than beef patties? The authors incorporate FS in CM prototypes, but lean beef has a less fat content. Thus, I suggest including the beef patties in the structuring and cooking comparison in Figure 5 and Table 1.

Minor comments:

1) It is mentioned that cells were seeded in the microcarrier for 8 days for expansion. From then, the time points of the cell culture process until cell aggregation in static culture (post-expansion) were not clear. How long was the duration of the whole study, from the cell seeding on edible microcarriers to lab-grown meat formation?

2) For Supplementary Figure 1b, the authors can consider changing the order of depiction of chondrogenesis, osteogenesis and adipogenesis based on the figure caption for consistency.

3) I am not sure if “maximal viabilities on day 8” (page 7) is the most appropriate phrase to use for the quantitative viability analysis as “maximal” implies that it has reached the highest possible point and the actual data should reflect a drop in cell viability beyond day 8. However, such conclusions cannot be made based on what is portrayed in Figure 2b and hence the term “highest” may be more appropriate compared to “maximal”.

- 4) The use of the word “interestingly” suggests that the results are surprising (page 7). Was the higher homogeneity in a dynamic culture not expected?
- 5) The authors may want to look through the manuscript, paying particular attention to the grammar/language used. For example, on page 7, authors can consider replacing “Along the bioreactor culture” with “Along with the bioreactor culture”.
- 6) In Figure 3b, the scale bar in the protein image has been cut off, so please check. Moreover, please put more details in the description of what each CLSM image is. For example, is the protein image taken from the chickpea protein sample?
- 7) The authors may want to check the manuscript for spelling errors. For example, on page 9, “approximately” is misspelt. “Conventional” is also misspelt on page 14.
- 8) The idea of using a plant-based fat substitute for cultured meat is interesting. It is understood that oleogel-based fat substitute supports the appearance and stiffness of the meat by providing solid properties by oleo gelation. However, the manuscript has acknowledged a difference in textural parameters of the FS with conventional beef fat. Since meat flavour is a vital factor for the palatability and acceptability of meat by consumers, I would hence like to question if the use of FS affects the meat flavour and other sensory attributes of produced meat.
- 9) There is a mistake in the figure number referenced on page 12. I believe the authors are referring to Figure 4e as opposed to Figure 3e. Please check the entire manuscript to ensure the correct figures are being referenced.
- 10) Please check the consistency of the decimal point presented in Supplementary Figure 6b.
- 11) What was the purpose of seeding the cells before transferring them to a spinner flask? Ideally, the cells can attach to the microcarriers in the spinner flask.
- 12) The authors should add the source of reagents. For example, on page 29, who is the supplier for Nile red and Nile blue reagents? Please add the details.
- 13) The cell culture part failed to provide the muscle cell culture details (expression of myosin heavy chain, desmin, and myogenin) - Immunofluorescence and qPCR.
- 14) One of the major limitations of lab-grown meat is mass production. The manuscript has placed a strong emphasis on the nutrition and sensory properties of the prototypes. The authors can consider commenting more on scaling up the potential of this cultured meat to highlight the impact of this research.
- 15) Were there any studies done investigating the shelf-life of produced lab-grown meat prototypes before and after cooking?

Reviewer #3 (Remarks to the Author):

This manuscript deals with using edible microcarriers in scalable bioreactor system to produce meat. The concept is interesting. However, some questions need to be addressed.

For Figure 2 and the related paragraph, some terms are confusing. For example, Figure 2b “Relative

cell viability" can be replaced by "Fold expansion". Viability is generally used in describing the ratio of live and dead cells. In addition, from the Figure 2b growth curve, it's hard to tell the cells reach their maximal expansion and there is still some empty space on microcarriers in Figure 1a.

On page 6, the author mentioned that the 50% medium exchange on days 3, 5, and 7 timeline was optimized. Is any data showing that even more aggressive medium exchange strategies (i.e., higher volume and more frequently) did not increase cell growth rate?

On page 7, cell expansions within the suspension plate, spinner flask, and bioreactor have the specific growth rates at 0.22, 0.16, 0.19 day⁻¹, respectively. Then, the authors mentioned that "These results were further validated through nuclei count, showing a similar amount of cell numbers per microtissue in all culturing conditions (Fig. 2c)." However, in Figure 3c, the lowest cell number is the bioreactor condition.

On page 23, "Edible cell microcarriers were seeded with the cells at a density of 10,000 cells/cm² in 100 mm suspension plates" which is a little confused. I assumed the density is calculated by cells per microcarrier surface, not cells per suspension plate surface. The authors can clarify it. I think the better way is to put the microcarrier amount and the cell number (equivalent to XXX cells/cm²) per each 100 mm suspension plate.

On page 24, the authors mentioned the pH sensor was calibrated by 2-point calibration. How about the DO sensor?

For bioreactor system, are there any records for pH, DO, temperature during the 8-day expansion period? The authors mentioned the air was pumped into the headspace in bioreactor. What is the pumping flow rate? The gas is air or O₂? If the pH is set to 7.2, how to control the pH? Some details should be included.

Response letter to reviewers' comments

We would like to thank all referees for their thorough and critical reviews, and the editor for her positive view of the manuscript, allowing us to improve our work and re-submit the current version. We have addressed all the reviewer's comments and performed new experiments, among them the fatty acid profile of the cultured meat in comparison to beef patties (new Supplementary Figure 7) and textural properties of beef patties that were determined and added to Table 1 for comparison with the newly developed CM products. We also completed an elemental analysis of decellularized micro-carriers and added the information to Figure 4e.

The major changes to the manuscript are highlighted. Our detailed response is enclosed below.

Reviewer 1:

I would like to see some aspects regarding nutrition addressed in the work since, as the authors pointed out, these approaches could be important for the tailoring aspect of the produced systems not only concerning the mechanical and texture properties but also for the nutritional aspects as well.

Following the reviewer's comment, the nutritional aspects were emphasized in the manuscript (lines 251-255, 295-298, 367-369, 371-374, 443-446). Furthermore, to better understand the nutritional composition of the developed CM, we analyzed the fatty acids profile of the FS and the CM in comparison to beef (Supplementary Figure 7).

Abstract

In my opinion, lacks objectivity and should be improved in that aspect. The main findings should be highlighted and well described (it seems a collection of bullet points).

We thank the reviewer for this comment. We thoroughly revised the abstract to be more objective and to highlight the main findings of the work, while complying with the formatting requirements.

Introduction

"...alternative to..."

Revised (line 35)

Can the authors elaborate on this

"CM product to enhance meat-like appearance, taste, and nutritional values without requiring harvesting steps." In what way the nutritional values are affected and the feasibility of it.

The nutritional values, appearance, and taste are affected by the choice of material from which the microcarriers are produced. As the microcarriers are integrated into the CM product (following their cellularization), their composition can be used as a mean to introduce nutrients such as proteins, oil, and fibers. In addition, nutraceuticals may be encapsulated in the microcarriers. This point was further elaborated in the manuscript (lines 54-57).

When confluency is reached, in what way the reactor volume limits the yield of the whole process? There is the possibility to promote confluency in several layers?

While some types of cells can keep proliferating when reaching confluency, and generate 2-3 cell layers, most primary cells including bMSC will stop proliferating when reaching confluency, as demonstrated in the attached figure below. We chose not to include these results in the manuscript since they were obtained using different culture conditions than the ones we present in the paper.

The reactor volume, on the other hand, limits the number of microcarriers that can be used per a single run and, consequently, the number of cells that can be expanded. When establishing an industrial production line, this factor can be optimized.

Why not consider adipocyte growth as a complement for fat production?

Adipocyte growth is another possible approach that can contribute to the fat mouthfeel and nutritional values. In the present work, however, we chose to develop a fat substitute due to its low cost, low saturated fat, and controllable properties.

Results

Is Minibio a trademark? If so please revise the wording.

Minibio is not a trademark. We revised the manuscript to clarify this is a specific instrument (lines 107, 511-512).

The foaming was a sole consequence of using the sparger or the agitation was responsible for it? Would a bubble column type-reactor with a bottom gas (porous or needle) sparger work? That would possibly avoid the microcarriers aggregation and foster a good enough gas dispersibility.

The foaming was the sole consequence of using the sparger, as we could avoid it using a gas overlay with the same agitation. A bubble column type-reactor with a bottom gas sparger could also work with no significant disadvantages or advantages over the current system.

Figure 2 should be revised to improve readability. Font size must be increased, as well as micrographs size. There is no readability in Figure 2F. The authors should consider dividing the image.

According to the reviewer's comment, Figure 2 was revised.

Was there any statistical difference among the cell number when the culture was performed under different conditions?

There was no statistical difference among the cell number under different conditions, marked as NS in Figure 2c. Following the reviewer's comment, we further detailed it in the figure legend.

"To grant CM with the sensorial and nutritional properties attributed to the fat component of animal-derived meat, an oleogel-based fat substitute (FS) was developed." this sentence is confusing to me. Please rewrite it.

The sentence was revised (lines 181-183).

It is not clear why TG was needed to produce CM prototypes. Can the author elaborate on that?

Transglutaminase crosslinking was necessary to generate cohesive structures from the microtissues and the oleogel-based fat substitute. Control samples without the addition of TG possessed a poor structure. This enzyme is approved as a food additive and as a

processing aid, and it is commonly used in the production of diverse foods including meat products. This issue was better explained in the introduction (lines 73-76) and the results (lines 238-239, 269-271) sections.

In figure 4A, the addition of a scale bar in the aggregates pictures would be helpful.

Scale bars were added to Figure 4A.

Correct the yy axis range in Figure 5d. The way it is portrayed (starting at 40%) is misleading and incorrect.

Figure 5d was amended accordingly.

What was the fat content of lean beef?

The fat content of lean beef was 4.3%. This information was also added to the manuscript (line 606).

Why there is no data on the textural properties of beef?

Textural data on beef was added to Table 1.

Why microcarriers can contribute to the sensorial and nutritional properties?

The microcarriers can contribute to the sensorial and nutritional properties by the choice of materials from which the microcarriers are produced as well as their production process (e.g. crosslinking, size, etc.). As the microcarriers are integrated into the CM product (following their cellularization), their mechanical and structural properties highly affect the CM's texture and mouthfeel, and their components affect CM's nutritional values, taste, and odor.

Aiming at increasing the scale of microcarrier production, are expected major cellular changes or even aggregation-related problems?

Microcarriers are produced using electrospray technology, which is scalable and is used in many food-tech and pharma industrial processes. As in the scale-up of any technological process, some adjustments may be required but we believe no major cellular changes or aggregation problems are expected.

"Hence, strong intermolecular interactions between the proteins in the oleogel-based..." substitute "strongly" with another word to avoid repetition.

The sentence was revised (line 390).

Any thoughts on how to tackle the "dark coloration" issue?

The color of the raw and cooked products can be controlled using different edible colorants, considering their browning reactions under heat. From our perspective, natural ones based on curcumin, beetroot, carotenes, etc. are preferred. Nevertheless, optimization of coloring can only take place once the final culture conditions are

achieved, including animal-free serum and supplements, as these greatly affect the product's color.

Fatty acid composition data for the CM prototypes would be a relevant result that would be beneficial to the overall discussion. Is it possible to add that?

We thank the reviewer for this excellent suggestion. We extracted the fat from the CM and the beef patties using the Soxhlet method, esterified the fatty acids, and analyzed them by gas chromatography with a flame ionization detector. The chromatograms and fatty acid profile were added as a new figure (Supplementary Figure 7). The results are discussed in the section on cultured meat prototypes: Structuring and characterization (lines 295-298, and 367-369).

Methods

In the oleogel production description, the first sentence (regarding lyophilisation) must come at the end of the paragraph.

The first sentence introduces the known method used for FS production, "oleogel-in-water emulsion-based template". Then, the specific information regarding the production process we used in our work is detailed. To make sure this is clear to the reader, we revised the sentence (lines 573-574).

Briefly describe how moisture content was determined.

The description was added (lines 610-614).

For textural tests define the probe dimensions as well as the dimensions of the tested samples.

The information was added (lines 628-633).

Did the authors perform some cutting tests? That would be relevant for a better understanding of the internal structuring of the prototypes.

We agree with the reviewer that a cutting test can further contribute to the textural analyses. Nevertheless, as the corresponding accessory is not available in our institute, we did not conduct the test. We do plan to purchase the cutting geometry for our following studies. At this stage of preliminary prototypes, we believe the current texture analyses make a good start in revealing the potential as well as the weaknesses of the developed prototypes.

Reviewer 2:

Although chickpea protein and canola oil are non-animal based, chitosan and collagen are still derived from animal sources. They are still being used to make edible micro-

carriers which is contradictory. Regardless, the paper is a proof of concept for using a combination of micro-carriers and FS for manufacturing cultured meat.

We agree with the reviewer that the use of animal-derived materials for cultured meat involves sustainability issues. As the reviewer pointed out, we used these materials in the current work to provide a proof of concept for the suggested platform. It is important for us to mention, though, that in our following work we focus on improving the system's sustainability, thus relying solely on plant-derived materials.

Major comments:

1) Interestingly, the authors are able to differentiate bMSCs in three different cells, which are chondrocytes, adipocytes, and osteocytes. However, the dominant cells in cultured meat (CM) should be myocytes (muscle cells). The protein in CM should come from the differentiated myocytes, which are myoblasts. It is well known that the differentiation of myoblasts into myotubes and, followingly, myofibers from MSCs is essential to obtain the desired texture of the final culture meat product. Could authors deliberate more on why the authors did not try to differentiate bMSCs into myocytes or do any analysis of this cell type?

We agree with the reviewer that the ideal CM should include muscle cells as the dominant cell type. Nevertheless, since our aim in the current work was to provide a proof of concept for using a combination of microcarriers and FS for cultured meat manufacturing, cell differentiation was not included in its scope. We do focus on myocytes-based CM in our following research projects, which we hope to complete in the upcoming year.

2) The paper mainly focused on the food science/processing and cooking aspects and failed to represent the molecular aspects of muscle differentiation during the development of the prototype.

We agree with the reviewer's comment. Our aim in the current work, however, was to provide a proof of concept for using a combination of microcarriers and FS for cultured meat manufacturing, as such a platform was never reported in the literature. We, therefore, did not focus on the differentiation of the cells on the microcarriers, though this is an important issue that we are addressing in one of our current research projects.

3) Elemental analysis may not be the most appropriate for understanding the nutritional composition of the fat substitutes. Could the authors elaborate more on the choice of the method instead of considering standardised nutritional analysis methods?

We agree with the reviewer that elemental analysis is not enough to gain a full understanding of the nutritional composition of the fat substitute. We used elemental analysis mainly to determine the nitrogen content of the products and from that calculate the protein content of cultured meat¹. It is useful for cases in which small

samples are available. Nevertheless, the composition of the FS is very clear from the well-defined components used for its production: 68% canola oil, 15% chickpea protein, and 17% GMS. This information was added to the manuscript (lines 600-601), while the elemental analysis was removed from Figure 3 to prevent confusion. To better understand the nutritional composition of the fat component of the developed CM, however, we analyzed the fatty acids profile of the FS and the CM in comparison to beef. This was done using Soxhlet extraction of the fat, methylation, and analysis of the fatty acid methyl esters using GC-FID, which is the official and common method to evaluate the nutritional composition of fats in food products. The results were added as Supplementary Figure 7.

4) On page 13, Figure 4e: The authors performed an elemental analysis of the cellularized microtissues and disc-like aggregates. However, it is necessary to compare this with decellularized microtissues. This is because the edible micro-carriers also contain the protein element from collagen. Thus, N and S elements which are present more in protein than fats, may come from the micro-carriers themselves but not from the cells since bMSC cells have not differentiated into muscle cells.

We thank the reviewer for this comment. We performed an elemental analysis of the pure microcarriers and added the results to Figure 4e (lines 251-255).

5) Why did the authors compare both CM prototypes with lean beef rather than beef patties? The authors incorporate FS in CM prototypes, but lean beef has a less fat content. Thus, I suggest including the beef patties in the structuring and cooking comparison in Figure 5 and Table 1.

We thank the reviewer for this comment. We have mistakenly labelled the specimens as “lean beef”, but they were actually patties produced from lean beef. We, therefore, revised the methods section to include the correct preparation and source details (lines 606-608), and we changed the text terminology and labels in the graphs. Furthermore, we characterized these patties and added the data to the revised Table 1.

Minor comments:

1) It is mentioned that cells were seeded in the microcarrier for 8 days for expansion. From then, the time points of the cell culture process until cell aggregation in static culture (post-expansion) were not clear. How long was the duration of the whole study, from the cell seeding on edible microcarriers to lab-grown meat formation?

The timeline for the production of the two prototypes was as detailed in the subsequent table, which was also added to the supplementary material to further clarify the production timeline. (Supplementary Table 1)

	Cell seeding	Cell expansion	Aggregation	Structuring	Total
Burger-like CM prototype	24hr	8 days	-	7 days	16 days
Layered CM prototype (disc-like)	24hr	8 days	7 days	7 days	23 days

2) For Supplementary Figure 1b, the authors can consider changing the order of depiction of chondrogenesis, osteogenesis and adipogenesis based on the figure caption for consistency.

According to the reviewer's comment, the figure legend was revised.

3) I am not sure if “maximal viabilities on day 8” (page 7) is the most appropriate phrase to use for the quantitative viability analysis as “maximal” implies that it has reached the highest possible point and the actual data should reflect a drop in cell viability beyond day 8. However, such conclusions cannot be made based on what is portrayed in Figure 2b and hence the term “highest” may be more appropriate compared to “maximal”.

The word “maximal” was replaced with “highest” (lines 135-137, 246-247).

4) The use of the word “interestingly” suggests that the results are surprising (page 7). Was the higher homogeneity in a dynamic culture not expected?

Thanks for the comment. The word “interestingly” was replaced with “notably”, which does not imply the results are surprising (line 144).

5) The authors may want to look through the manuscript, paying particular attention to the grammar/language used. For example, on page 7, authors can consider replacing “Along the bioreactor culture” with “Along with the bioreactor culture”.

We thank the reviewer for drawing our attention to this problem. The entire manuscript text was screened, and grammar/language errors were fixed.

6) In Figure 3b, the scale bar in the protein image has been cut off, so please check. Moreover, please put more details in the description of what each CLSM image is. For example, is the protein image taken from the chickpea protein sample?

The scale bar was corrected, and the legend was revised to include additional details.

7) The authors may want to check the manuscript for spelling errors. For example, on page 9, “approximately” is misspelt. “Conventional” is also misspelt on page 14.

We thank the reviewer for drawing our attention to these mistakes. The entire manuscript text was proofread, and spelling errors were fixed.

8) The idea of using a plant-based fat substitute for cultured meat is interesting. It is understood that oleogel-based fat substitute supports the appearance and stiffness of the meat by providing solid properties by oleo gelation. However, the manuscript has acknowledged a difference in textural parameters of the FS with conventional beef fat. Since meat flavour is a vital factor for the palatability and acceptability of meat by

consumers, I would hence like to question if the use of FS affects the meat flavour and other sensory attributes of produced meat.

This is an important question because there is no doubt that the FS will affect all the sensorial properties of the developed cultured meat. Nevertheless, at the current stage of the research, we cannot address these issues since the culture conditions are not suitable for consumption as food (e.g., culture media, serum, growth factors, etc.) and, therefore, it is not safe to taste the produced CM and evaluate its flavor. The sensorial attributes of CM, and particularly its FS, will indeed be optimized once it is prepared in food-compatible conditions. For instance, the sensorial properties can be tailored by choosing a specific oil type along with a specific oleogelator, that can also serve as a flavor delivery system. In addition, the choice of protein type and its adequate processing can significantly contribute to the FS sensorial attributes. It is important to note, though, that plant oil-based fat substitutes are widely used in the food industry without negative effects on flavor, odor, and other sensorial attributes.

9) There is a mistake in the figure number referenced on page 12. I believe the authors are referring to Figure 4e as opposed to Figure 3e. Please check the entire manuscript to ensure the correct figures are being referenced.

The mistake was fixed, and the entire manuscript was checked to ensure the correct figures are being referenced.

10) Please check the consistency of the decimal point presented in Supplementary Figure 6b.

Figure 6b was amended according to the reviewer's comment.

11) What was the purpose of seeding the cells before transferring them to a spinner flask? Ideally, the cells can attach to the microcarriers in the spinner flask.

We agree with the reviewer that cell seeding in the spinner flask will be more commercially-viable, and in our following research projects we have optimized the seeding in the spinner flask. However, in the experiments presented in the current work, static seeding was applied due to its simplicity.

12) The authors should add the source of reagents. For example, on page 29, who is the supplier for Nile red and Nile blue reagents? Please add the details.

Details were added according to the reviewer's comment (lines 482, 484-487, 546, 548-549, 671-672, etc.).

13) The cell culture part failed to provide the muscle cell culture details (expression of myosin heavy chain, desmin, and myogenin) - Immunofluorescence and qPCR.

We agree with the reviewer that the ideal CM should include muscle cells as the dominant cell type. Nevertheless, since our aim in the current work was to provide a proof of concept for using a combination of microcarriers and FS for cultured meat manufacturing, cell differentiation was not included in its scope. We, therefore, cultured

the cells under conditions that maintain their stemness and we show in Figure 2 using both immunofluorescence staining and flow cytometry, that the cells indeed remain undifferentiated. In our subsequent research projects, we focus on cell differentiation following their expansion on edible microcarriers.

14) One of the major limitations of lab-grown meat is mass production. The manuscript has placed a strong emphasis on the nutrition and sensory properties of the prototypes. The authors can consider commenting more on scaling up the potential of this cultured meat to highlight the impact of this research.

We revised the manuscript to better emphasize the scaling-up options and limitations of our platform (Abstract, introduction, discussion, conclusions).

15) Were there any studies done investigating the shelf-life of produced lab-grown meat prototypes before and after cooking?

At this stage of the research, we did not evaluate the shelf-life of the CM prototypes. Shelf life, however, is a major factor in the success of any CM production and it will be thoroughly addressed in our subsequent research, once reaching food-compatible culture conditions.

Reviewer 3:

For Figure 2 and the related paragraph, some terms are confusing. For example, Figure 2b “Relative cell viability” can be replaced by “Fold expansion”. Viability is generally used in describing the ratio of live and dead cells.

Following the reviewer’s comment, we replaced the term “Relative cell viability” to “Fold expansion” in Figure 2 and Figure 4 and their captions.

In addition, from the Figure 2b growth curve, it’s hard to tell the cells reach their maximal expansion and there is still some empty space on microcarriers in Figure 1a.

While in the images some empty space can still be seen, our preliminary experiments showed that viability levels decline on the 9th day of culture. We, therefore, chose to end the expansion after 8 days (please see the figure below). We chose not to include these results in the manuscript since they were obtained using different culture conditions than the ones we present in the paper.

On page 6, the author mentioned that the 50% medium exchange on days 3, 5, and 7 timeline was optimized. Is any data showing that even more aggressive medium exchange strategies (i.e., higher volume and more frequently) did not increase cell growth rate?

Aiming for a commercially-viable process, we wanted to apply less aggressive strategies and reduce the culture costs, while still achieving fast expansion. We, therefore, did not address more aggressive medium exchange strategies.

On page 7, cell expansions within the suspension plate, spinner flask, and bioreactor have the specific growth rates at 0.22, 0.16, 0.19 day⁻¹, respectively. Then, the authors mentioned that “These results were further validated through nuclei count, showing a similar amount of cell numbers per microtissue in all culturing conditions (Fig. 2c).” However, in Figure 3c, the lowest cell number is the bioreactor condition.

Figures 2b and 2c show the fold expansion of bMSCs and cell numbers, respectively. These results reveal no significant differences between the 3 culturing conditions. Similarly, the calculated specific growth rates were not significantly different. We, therefore, described the results as comparable or similar. To further clarify this, we amended the text (lines 139-142).

On page 23, “Edible cell microcarriers were seeded with the cells at a density of 10,000 cells/cm² in 100 mm suspension plates” which is a little confused. I assumed the density is calculated by cells per microcarrier surface, not cells per suspension plate surface. The authors can clarify it. I think the better way is to put the microcarrier amount and the cell number (equivalent to XXX cells/cm²) per each 100 mm suspension plate.

We thank the reviewer for drawing our attention to the unclear text. The cells were seeded on the microcarriers in 100 mm suspension plates, but the density of 10,000 cells/cm² refers to the surface area of the microcarriers, which was calculated as detailed in the methods section (line 492). To clarify this, we amended the text (lines 505-506).

On page 24, the authors mentioned the pH sensor was calibrated by 2-point calibration. How about the DO sensor?

After autoclaving, the dissolved oxygen (DO) sensor was calibrated using a 2-point method. The bioreactor was first aerated with the air through its headspace for 15 min to set the first calibration point as 100% DO. The bioreactor was then pumped with nitrogen for 15 min to remove all the oxygen from the culture medium and set the second calibration point as 0% DO. This information was added to the text (lines 516-522).

For bioreactor system, are there any records for pH, DO, temperature during the 8-day expansion period? The authors mentioned the air was pumped into the headspace in bioreactor. What is the pumping flow rate? The gas is air or O₂? If the pH is set to 7.2, how to control the pH? Some details should be included.

There are no records for pH, DO, and temperature during the expansion period but their values were automatically adjusted by the software to reach the set points. This information was added to the manuscript (lines 525-526). Unexpected events of offset values turn on an alarm. However, no such events happened during our experiments.

The air was pumped into the headspace in the bioreactor at 1 ml/min. This information was added to the manuscript (lines 522-523).

References:

1. S. Serrano, F. Rincón, J. García-Olmo, Cereal protein analysis via Dumas method: Standardization of a micro-method using the EuroVector Elemental Analyser, *Journal of Cereal Science*, Volume 58(1), 2013, Pages 31-36.

REVIEWERS' COMMENTS

Reviewer #1 (Remarks to the Author):

After reviewing the authors' rebuttal, where all the concerns raised by this Reviewer were properly addressed, my suggestion is that the revised version of the manuscript is suitable to be accepted for publication.

Reviewer #2 (Remarks to the Author):

Overall comments:

- Overall, the authors have acknowledged all comments. However, the main concern regarding the lack of muscle cell work remains unaddressed. Therefore, instead of stating that muscle differentiation will be the focus of future work, it may be beneficial to include some of that work in this paper.
- It is recommended that the authors thoroughly review the entire manuscript for grammatical and spelling errors. For example, in line 54, instead of "This approach can be also used...", it should be "This approach can also be used...". Additionally, "weighed" and "weighted" has vastly different meanings, and I believe in line 611, the authors are referring to the act of measurement, which is "weighed". "Prototypes" is also misspelt in line 611.
- Please check the spacing errors throughout the manuscript, as some errors have been found. For example, "500ml" on page 6, line 107 should be written as "500 ml".

Additional Comments:

- Line 35: The word "consumption" is unnecessary in this line. The authors can consider removing it to improve the overall readability of the manuscript.
- Table 1: Why were hardness, adhesiveness, cohesiveness and chewiness the only textural parameters presented in table 1? Why was springiness excluded from the results of the analysis despite it being described in detail in the method section of the manuscript?
- On page 8, lines 181-183, the authors stated that fat contributes to taste and mouthfeel, but mammalian fat is not considered a healthy or nutritious food. It would be helpful if the authors could provide references to support this statement.
- Regarding the elemental analysis, it would be useful to know whether the weight of all samples was the same before the test was run. Additionally, is it possible that the higher values for N, C, H, and S could be due to differences in the weight of the samples?
- Instead of using symbols, the authors should spell out " $\omega 3$ " as "Omega-3".

Reviewer #3 (Remarks to the Author):

Thank you very much for addressing all my questions.

Response letter to reviewers' comments

Reviewer #1 (Remarks to the Author):

After reviewing the authors' rebuttal, where all the concerns raised by this Reviewer were properly addressed, my suggestion is that the revised version of the manuscript is suitable to be accepted for publication.

We thank the reviewer for endorsing the manuscript.

Reviewer #2 (Remarks to the Author):

Overall comments:

- Overall, the authors have acknowledged all comments. However, the main concern regarding the lack of muscle cell work remains unaddressed. Therefore, instead of stating that muscle differentiation will be the focus of future work, it may be beneficial to include some of that work in this paper.

From our point of view, in the development of scaffolds for cultured meat, the research on muscle differentiation is an extremely important subject, requiring numerous analyses concerning many different aspects. These include, on the one hand, the primary ability to differentiate the cells, as well as the efficiency of differentiation at different culture timings and under diverse conditions, which are all dependent on the type of scaffold and its properties. On the other hand, the effect of cell differentiation on the cultured meat's attributes should be thoroughly investigated, including the texture and mouthfeel, nutritional values, the different post-culture processing technologies that can be applied, etc. As the scope of the current paper is providing a proof of concept for the use of cellularized microcarriers in combination with oleogel-based fat substitute as a platform for cultured meat, we respectfully disagree with the reviewer. We believe that including only some of the differentiation work would be irrelevant to the current work and disconnected from the comprehensive work focusing on muscle differentiation that we currently perform in the lab with diversified cultured meat scaffolds.

- It is recommended that the authors thoroughly review the entire manuscript for grammatical and spelling errors. For example, in line 54, instead of "This approach can be also used....", it should be "This approach can also be used...". Additionally, "weighed" and "weighted" has vastly different meanings, and I believe in line 611, the authors are referring to the act of measurement, which is "weighed". "Prototypes" is also misspelt in line 611.

Corrected.

• Please check the spacing errors throughout the manuscript, as some errors have been found. For example, “500ml” on page 6, line 107 should be written as “500 ml”.
Corrected.

Additional Comments:

Line 35: The word “consumption” is unnecessary in this line. The authors can consider removing it to improve the overall readability of the manuscript.

Corrected.

Table 1: Why were hardness, adhesiveness, cohesiveness and chewiness the only textural parameters presented in table 1? Why was springiness excluded from the results of the analysis despite it being described in detail in the method section of the manuscript?

We thank reviewer for this comment, springiness is now included in Table 1 and in the table presented in Supplementary Figure 4d. Although there is a vast of different textural parameters, hardness, adhesiveness, cohesiveness, chewiness and springiness are chosen as the most appropriate descriptive parameters for meat type samples. We avoided presenting a large number of the textural parameters that could be confusing to readers, but rather focused on the most relevant ones. Similar parameters were previously reported by numerous authors when examining meat and cultured meat samples:

- Tanaka, Ri., Sakaguchi, K., Yoshida, A. *et al.* Production of scaffold-free cell-based meat using cell sheet technology. *npj Sci Food* **6**, 41 (2022). <https://doi.org/10.1038/s41538-022-00155-1>
- Paredes, J., Cortizo-Lacalle, D., Imaz, A.M. *et al.* Application of texture analysis methods for the characterization of cultured meat. *Sci Rep* **12**, 3898 (2022). <https://doi.org/10.1038/s41598-022-07785-1>
- Yan-Yan Zheng, Hao-Zhe Zhu, Zhong-Yuan Wu, *et al.* Evaluation of the effect of smooth muscle cells on the quality of cultured meat in a model for cultured meat, *Food Research International* **150** (2021). <https://doi.org/10.1016/j.foodres.2021.110786>.

On page 8, lines 181-183, the authors stated that fat contributes to taste and mouthfeel, but mammalian fat is not considered a healthy or nutritious food. It would be helpful if the authors could provide references to support this statement.

We thank reviewer for this comment. The sentence was corrected to: “The fat component in animal-derived meat greatly contributes to its taste, mouthfeel, and texture” and a new reference was added:

Drewnowski, A., & Almiron-Roig, E. (2009). Human perceptions and preferences for fat-rich foods. In: *Fat detection: Taste, texture, and post ingestive effects*, 23, 265. <https://www.ncbi.nlm.nih.gov/books/NBK53528/#ch11.s2>

Regarding the elemental analysis, it would be useful to know whether the weight of all samples was the same before the test was run. Additionally, is it possible that the higher values for N, C, H, and S could be due to differences in the weight of the samples?

The weight of all samples was nearly the same (0.5 ± 0.1 g), thus this factor did not contribute to the results. The exact weight of each sample is recorded prior to measurement, and the results are directly expressed by the instrument as the percentage of each element. This information was added to the manuscript (lines 580-582).

Instead of using symbols, the authors should spell out “ ω_3 ” as “Omega-3”.

Corrected.

Reviewer #3 (Remarks to the Author):

Thank you very much for addressing all my questions.

We thank the reviewer for endorsing the manuscript.